

# Predicting and testing a gene network regulating seed germination in *Arabidopsis*

Ming Yang and Yixing Wang

Plant Biology, Ecology, and Evolution, Oklahoma State University, Stillwater, OK, United States

## ABSTRACT

Maternal auxin signaling inhibits seed germination in *Arabidopsis*, but little is known about the gene network that mediates the inhibition of seed germination by auxin. Based on publicly available data, we first determined the expression patterns of *AUXIN SIGNALING F-BOX* (*AFB*) genes in the funiculus (FUN)-chalazal seed coat (CSC) continuum. We found that *TIR1*, *AFB1*, and *AFB4* exhibited a down-expression gradient and *AFB2*, *AFB3*, and *AFB5* an up-expression gradient from FUN to CSC, and members in each of the two *AFB* groups were linearly correlated in expression in FUN, CSC, and the distal seed coat (DSC). We then searched for auxin-regulated genes that exhibited one of the two expression gradients. We found 118 such genes that were assigned into four groups based on their response mode to auxin and expression gradient direction. The four groups were further broken down into 12 subgroups based on the linear correlation coefficient values. Only three of the 12 subgroups, including a total of 30 genes with 21 of them being known or highly likely to function in the seed germination process. To explore whether additional genes of the remaining nine play roles in seed germination, we tested mutants of five of them in a germination assay and found all of them exhibited either delayed or hastened germination. The experimental results support the validity of our approach for predicting the involvement of these genes in seed germination. Based on publicly available data and data from this investigation, we constructed a gene network, which should provide a valuable framework and new clues for future studies of the molecular mechanism controlling seed germination.

## INTRODUCTION

A mature angiosperm seed has a seed coat that surrounds the endosperm that surrounds the embryo. The seed coat develops from maternal tissues that typically include the integuments and the chalaza of the ovule (*Chaudhury et al., 2001*; *Millar et al., 2015*). Regulation of seed germination can occur in any part of the seed, but the maternal control of seed germination by the seed coat is a major mechanism. Studies in diverse species have established that the seed coat exerts negative physiological effects on germination (*Kazachkova et al., 2016*; *Webster et al., 2016*; *Fedi et al., 2017*; *Bian et al., 2018*; *Shah et al., 2018*; *Wang et al., 2019*; *Zhang et al., 2020*). How such effects are established, maintained, and abolished in the seed coat is still poorly understood at the molecular level.

Corresponding author
Ming Yang, ming.yang@okstate.edu

In particular, limited knowledge is available on the structures of molecular networks in the seed coat that control seed germination.

Plant hormones play influential roles in seed germination. Abscisic acid (ABA) is known for promoting seed dormancy, whereas gibberellins (GAs) for breaking seed dormancy. The ratio between the concentrations of these two hormones in the seed is a major parameter for determining whether the seed is dormant or not. The ABA and GA signaling pathways interact with each other. The ABA pathway negatively regulates the GA pathway, and the GA pathway both negatively and positively regulates the ABA pathway (*Tuan et al., 2018*). An auxin signaling pathway also interacts with the ABA signaling pathway to promote seed dormancy (*Brady et al., 2003*; *Liu et al., 2007, 2013*; *Matilla, 2020*; *Pellizzaro et al., 2020*). A major theme here is that auxin signaling activates the functions of AUXIN RESPONSE FACTOR 10 and 16 (ARF10/16) that further induce ABA-INSENTITIVE 3 (ABI3) expression. ABI3 is a major component of the ABA signaling pathway for seed dormancy. It should be expected that a molecular network regulating seed germination should involve these three hormones and possibly other hormones. This network from here on is referred to as the seed germination network (SGN).

Auxin signaling in plants is mediated by the Skp1-Cullin-F-box protein (SCF) ubiquitin ligases. In Arabidopsis, the F-box proteins in the SCFs for auxin signaling are the AUXIN SIGNALING F-BOX (AFB) proteins. An AFB in the SCF complex binds both auxin and a substrate that typically is a transcriptional repressor in the Aux/IAA family (*Dharmasiri, Dharmasiri & Estelle, 2005*; *Kepinski & Leyser, 2005*; *Tan et al., 2007*; *Mockaitis & Estelle, 2008*; *Hayashi, 2012*). Binding of auxin by an AFB enhances the AFB's binding to the Aux/IAA protein (*Tan et al., 2007*). Upon such binding, the SCF ubiquitinates the Aux/IAA protein, leading to its degradation *via* the 26S proteasome pathway (*Skowyra et al., 1997*). Degradation of the Aux/IAA protein removes its inhibition on an ARF that is a transcription factor that either negatively or positively regulates the transcription of downstream genes (*Mockaitis & Estelle, 2008*; *Liscum & Reed, 2002*; *Okushima et al., 2005*; *Li et al., 2016*). There are six AFBs in Arabidopsis, and the functional redundancy among the AFBs seems moderate since the highly homologous TIR1 and AFB1 (70% identical at the amino acid level) are not functionally equivalent as demonstrated by a promoter swap experiment (*Parry et al., 2009*). Therefore, different AFBs, if included in the SGN, may play distinct roles. Indeed, we recently reported that neither AFB1 nor AFB5 was redundant in inhibiting seed germination and their mutations synergistically promoted seed germination in the double mutant when compared with the corresponding single mutants (*Wang et al., 2022*). Moreover, we found that the expression domains of *AFB1* and *AFB5* overlapped in the funiculus (FUN) in late seed development and in the hilum of the seed coat when the seed abscised from FUN, but only *AFB1* was transiently expressed in a small chalazal region of the seed coat that surrounded the hilum during imbibition (*Wang et al., 2022*).

Our experimental work above together with the transcriptomic data available for Arabidopsis provides an opportunity for exploring the structure of the SGN. Here we describe the process and outcome of developing the initial SGN, and the experimental

validation of some of the components of the SGN. This initial SGN may help fully elucidate the SGN in the future.

## MATERIALS AND METHODS

### A strategy for identifying separate groups of co-expressed auxin-responsive genes in the FUN-CSC continuum

The FUN and chalazal seed coat (CSC) form a tissue continuum in seed development. This FUN-CSC continuum is the only route for directly supplying nutrients and other molecules from the maternal tissues to the developing seed. Conceivably, the genes expressed in the FUN-CSC continuum can exert maternal effects on seed development and germination. Transcriptomic datasets for genes in parts of the Arabidopsis seed including FUN, CSC, and distal seed coat (DSC) at different developmental stages are available from *Belmonte et al. (2013)* and *Khan et al. (2015)*. These datasets were conveniently compiled in a single Excel file by Khan et al. (Dataset S1 in *Khan et al. (2015)*). Moreover, datasets of auxin-up- and-downregulated genes are available from microarray studies (*Goda et al., 2004*). It is possible to identify components of auxin signaling networks in the FUN-CSC continuum by searching for genes present in both the FUN and CSC gene expression datasets and the auxin-regulated gene datasets. The expression trends (upward or downward from FUN to CSC) of these genes can also be compared with the expression trend(s) of the *AFB* genes. If the *AFB*s exhibit different expression trends in the FUN-CSC continuum, the identified auxin-regulated genes can be further assigned to separate groups based on their expression trends in the FUN-CSC continuum and their modes of response (up or down) to auxin. In particular, Dataset S1 in *Khan et al. (2015)* were downloaded and reduced to contain only the raw microarray signals of the genes expressed in FUN, CSC (corresponding to CZSC in *Khan et al. (2015)*), and DSC (corresponding to SC in *Khan et al. (2015)*) at the mature-green-cotyledon stage. The auxin-regulated genes in this reduced dataset were identified by manually searching for the auxin-regulated loci numbers reported in *Goda et al. (2004)*. These auxin-regulated genes were further manually grouped based on whether they are down- or upregulated by auxin and whether they exhibit an up or a down expression pattern from FUN to CSC, as shown in Table S1. The grouping of *AFB*s was also conducted based on their up or down expression pattern from FUN to CSC (Table S1). When developing the gene groups in Table S1, genes whose raw microarray signal values in both FUN and CSC are less than 10 were omitted to reduce potential false positives in this investigation.

### Linear regression analysis for assigning subgroups of co-expressed auxin-responsive genes in the funiculus and seed coat

The available gene expression data contain microarray signal values for a gene expressed in the FUN, CSC, and DSC regions. The three data points from FUN, CSC, and DSC should enable a linear regression analysis of *in vivo* mRNA concentrations between gene pairs after proper normalizations, which may provide additional evidence for assigning genes in the above-mentioned groups to subgroups. Assuming that the RNA isolation efficiency was the same for the FUN, CSC, and DSC tissues, the relative *in vivo* concentration of an

mRNA can be defined by its microarray signal/its tissue volume. Because the tissues collected for the RNA isolation were from standard sections that had a fixed thickness, the volume ratios between any two of the FUN, CSC, and DSC regions are approximately equal to their area ratios. Based on the images provided by *Belmonte et al. (2013)*, the areas of the FUN and CSC were approximately the same and the area of the DSC was approximately 26.8 times of that of FUN or CSC (areas were measured using SigmaScan 5.0, Systat Software). Therefore, we normalized the microarray signal values for DSC by dividing them by a factor of 26.8 (Table S1). We then conducted linear correlation analysis using the normalized values along with the original FUN and CSC values in Excel. We used a range of high coefficient values, $1 \geq R^2 \geq 0.90$, to determine whether two genes belong to the same subgroup.

## Confirmation of T-DNA insertion mutants

A total of 10 mutants at five loci (two at each locus) of the identified genes without a known function in seed germination, all in the Columbia (Col-0) accession, were obtained from the Arabidopsis Biological Resource Center at The Ohio State University. These were T-DNA-insertion mutants that were verified by polymerase chain reactions (PCR) using their genomic DNAs as templates and T-DNA and gene-specific primers and a routine PCR protocol on a thermocycler (Tables S2 and S3; Fig. S1). Transcription of these genes in the corresponding mutants and Col-0 was investigated by reverse transcription (RT)-PCR with a primer in the T-DNA insertion and a gene-specific primer (Tables S3 and S4) and cDNA samples synthesized using Thermo Scientific Verso cDNA synthesis Kit (Thermo Fisher Scientific, Waltham, MA, U.S.A.) and RNA samples extracted from inflorescences using RNeasy Plant Mini Kit (Qiagen, Valencia, CA, U.S.A.).

## Testing seed germination in mutants of genes predicted to function in seed germination

The seeds for testing germination of a mutant line and the wild-type control Col-0 were harvested from plants that were grown at approximately the same time (planted on the same date and harvested zero to a few days apart) in the same growth chamber as previously described (*Wang et al., 2022*). The mutant and Col-0 seeds were air-dried at the same location and room temperature for the same duration (5–6 weeks) and were stored at the same location and room temperature in 1.5 mL cryogenic tubes with the screw cap tightened. At the time of the germination experiment, the seeds were less than a year old from the date of harvesting. Germination tests were conducted similarly to those previously described (*Wang et al., 2022*). Specifically, each piece of filter paper (9 cm in diameter) was placed in the lid of a petri dish (10 cm × 1.5 cm) and wetted with 2 mL double-deionized water. More than 100 seeds of each genotype were sprinkled onto the filter paper and covered with the bottom of the petri dish. The petri dishes were incubated upside down in a growth chamber at approximately 22 °C with approximately 100 μmol photons $m^{-2}s^{-1}$ in light intensity and a 16 h-light/8 h-darkness regime. Seeds were scored for germination under a dissecting microscope at the time of 50-h imbibition. A seed was considered germinated if a region of the seed coat was broken. If the germination status

could not be assessed due the downfacing of the radicle, the seed was flipped over with a syringe needle to determine its germination status. More than 100 seeds in each petri dish were scored to determine the germination frequency. The germination frequency of each genotype was determined eight or 12 times in separate petri dishes. For each mutant petri dish, one Col-0 petri dish was included as control; they were placed as one cluster of petri dishes in the growth chamber. For minimizing the difference in incubation time between the petri dishes due to the time spent in the seed counting process, four or fewer mutant-wild-type pairs of petri dishes were investigated in a single experiment. These measures should minimize the confounding effect of variations in seed germination between experiments at different times, and enable paired *t*-test between the mutant and corresponding wild-type samples.

## Constructing a gene network for controlling seed germination

Three groups of highly coexpressed genes identified earlier (Table S1) included components in the auxin, ABA, and brassinosteroid signaling pathways that are known to act in the seed germination process. Therefore, it is possible that all genes in the three groups act in a gene network that regulates seed germination by integrating the hormonal signaling processes. To explore such a possibility, datasets of genes whose expression is regulated by the three hormones were downloaded from the supplementary data linked to the publications (*Goda et al., 2004*; *Okamoto et al., 2010*; *Tian et al., 2020*; *Liu et al., 2020*). The potentially seed-germination-related genes in the three groups identified in this investigation were manually searched in these datasets. These searches revealed in-group and between-group connections for the genes in the three groups, enabling the construction of a gene network. This network was first constructed in Cytoscape_3.9.1 (Fig. S2 and Data S1–S3) and then, for simplicity, in Microsoft PowerPoint. This network contains 30 identified genes in the three group, the six *AFB*s, *BES1* and *BZR1* involved in brassinosteroid signaling, and *ABI3*, *4*, and *5* involved in ABA signaling.

# RESULTS

## Identification of auxin-down- or upregulated coexpressed genes in FUN and CSC

Following the strategy described in Materials and Methods, *in silico* searches uncovered that *TIR1*, *AFB1*, and *AFB4*, (the *AFB1* group) were expressed in a downward trend (Fig. 1A) whereas *AFB2*, *AFB3*, and *AFB5* (the *AFB5* group) in an upward trend (Fig. 1B) from FUN to CSC in the dataset generated by *Belmonte et al. (2013)* and *Khan et al. (2015)*. For all *AFB*s, the expression values of the distal seed coat (DSC), after normalization (see Materials and Methods for the rational for the normalization), were small compared to those of FUN and CSC (Fig. 1). Moreover, 118 genes, either down- or upregulated by auxin according to *Goda et al. (2004)*, were also identified in the same dataset, and were assigned into four groups based on whether they had a downward or an upward expression trend from FUN to CSC and whether they were down- or upregulated by auxin (Table S1).
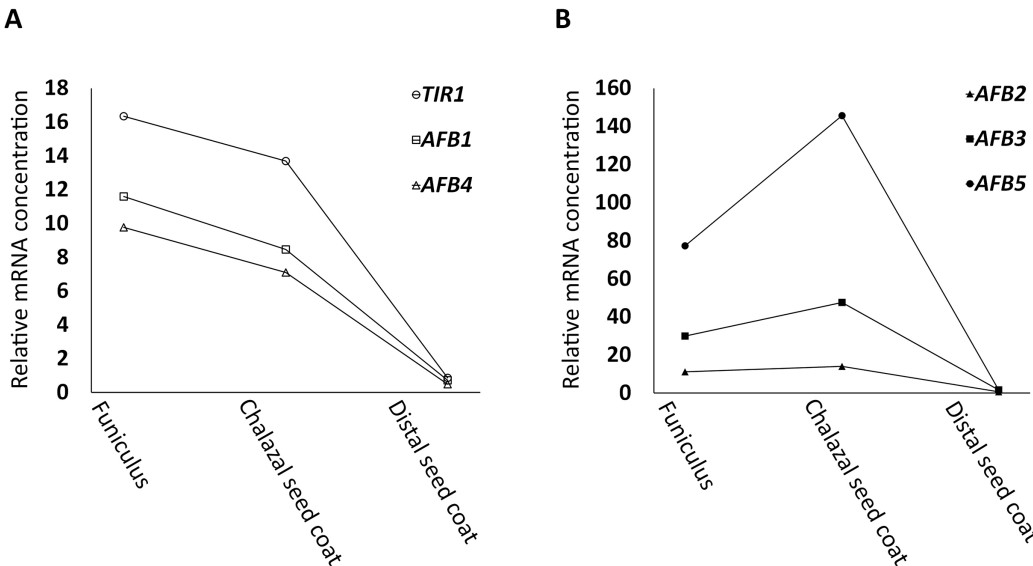

**Figure 1 Relative mRNA concentrations of *AFBs* in FUN, CSC, and DSC of nearly mature seeds.** (A) *TIR1*, *AFB1*, and *AFB4* exhibit a trend of downregulation from FUN to CSC. (B) *AFB2*, *AFB3*, and *AFB5* exhibit a trend of upregulation from FUN to CSC. The microarray signal levels reported by *Belmonte et al. (2013)* and *Khan et al. (2015)* were used as the relative mRNA concentrations, except for the values of DSC, which were normalized as described in Materials and Methods.

## Identification of genes that are highly correlated in expression with genes known to be involved in seed germination

To further investigate how the *AFBs* in the *AFB1* group and the *AFB5* group are correlated in expression, respectively, linear correlation analysis was conducted as described in Materials and Methods. The three *AFBs* in the *AFB1* group were found to be highly correlated ($R^2$ = 0.99, *TIR1 vs. AFB1*; $R^2$ = 1, *AFB4 vs. AFB1*; Fig. 2A), and so were the *AFBs* in the *AFB5* group ($R^2$ = 0.99, *AFB3 vs. AFB5*; $R^2$ = 0.92, *AFB2 vs. AFB5*; Fig. 2B). However, the $R^2$ value between *AFB1* and *AFB5* was only 0.57. These relationships among the members within the *AFB1* group or the *AFB5* group, and between *AFB1* and *AFB5* are consistent with the *AFB1* group and the *AFB5* group exhibiting opposite trends in expression in the FUN-CSC continuum.

The same linear correlation analysis was also carried out with the four groups of genes in Table S1. Based on the $R^2$ values, Group 1 was further divided into two subgroups, Group 1A and 1B, Group 2 into three subgroups, Group 2A–C, Group 3 into three subgroups, Group 3A–C, and Group 4 into four subgroups, Group 4A–D (Table S1). As an example for demonstrating that the genes within a subgroup were highly correlated, the sums of the expression levels of all genes in FUN, CSC, and DSC in Group 1A, excluding those of a member of the group, *At2g28470* (arbitrarily chosen), were plotted against those of *At2g28470*, which exhibited a linear correlation with $R^2$ = 0.98 (Fig. 3A). A similar level of linear correlation was found between the sum of Group 1B excluding *At4g20460* (arbitrarily chosen) and *At4g20460* with $R^2$ = 0.99 (Fig. 3B). These two group values, on the

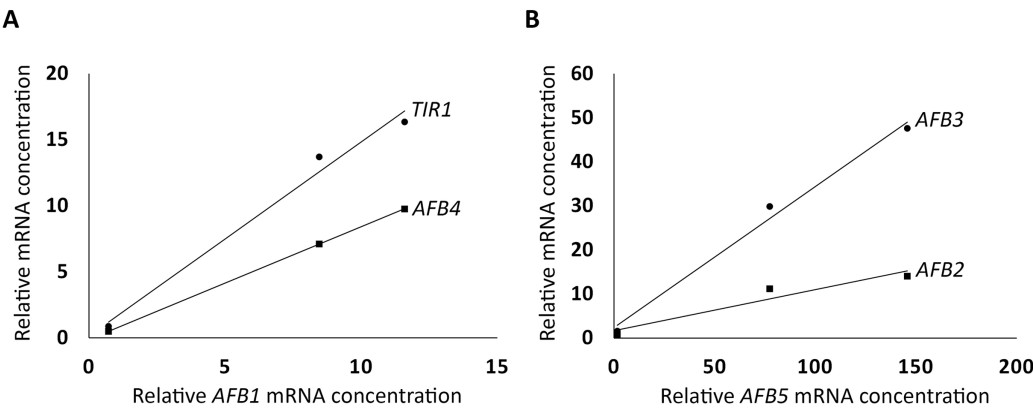

**Figure 2 Linear correlations in relative mRNA concentrations in FUN, CSC, and DSC among *TIR1*, *AFB1*, and *AFB4*, and among *AFB2*, *AFB3*, and *AFB5*.** (A) *TIR1* and *AFB4* were plotted against *AFB1*, respectively. (B) *AFB2* and *AFB4* were plotted against AFB5, respectively.

other hand, had $R^2 = 0.59$ when plotted against each other (Fig. 3C). These results indicated that Group 1A and 1B were separate groups, suggesting that they were not regulated by the same set of transcriptional factors. As conducted with Group 1A and 1B, the separation of Group 4A from Group 4B is shown in Fig. 4, with the intragroup $R^2$ values of 0.98 and 0.95, respectively, and the intergroup $R^2 = 0.86$ (Fig. 4). Among the 12 subgroups identified, only Group 1A, 1B, and 4A were found to contain genes that are known or highly likely to function in the seed germination process (Tables 1 and 2). In particular, Group 1A and 1B (Table 1) and Group 4A (Table 2) contained five, seven, and nine genes apparently functioning in the seed germination process, respectively. Nine genes in these three subgroups were unknown for a role in seed germination, but their close associations with the other members in the subgroups raised the possibility that they also participate in the seed germination process. Auxin should downregulate the genes in Group 1A and 1B and upregulate the genes in Group 4A. Furthermore, the expression trends of these genes and the *AFBs* in the FUN-CSC continuum suggest that the *AFB1* group downregulates Group 1A and 1B and the *AFB5* group upregulates Group 4A. It is noted here that the statistical *p* values of the above correlations are not provided because each of the variables only had three data points that were too few for producing meaningful *p* values.

## Experimental confirmation of the functions of five newly identified genes in seed germination

To experimentally test whether the genes unknown for their involvement in seed germination in Tables 1 and 2 indeed function in seed germination, we attempted to obtain two T-DNA insertion mutant alleles for each of them. We were able to confirm two homozygous mutant alleles for each of the five loci that included *At1g78090*, *At4g35060* (*HMP39*), *At1g51170*, *At3g13380* (*BRL3*), and *At2g23060*. One or two mRNA transcripts that contained a part of the T-DNA insertion were detected in each of the mutant alleles by
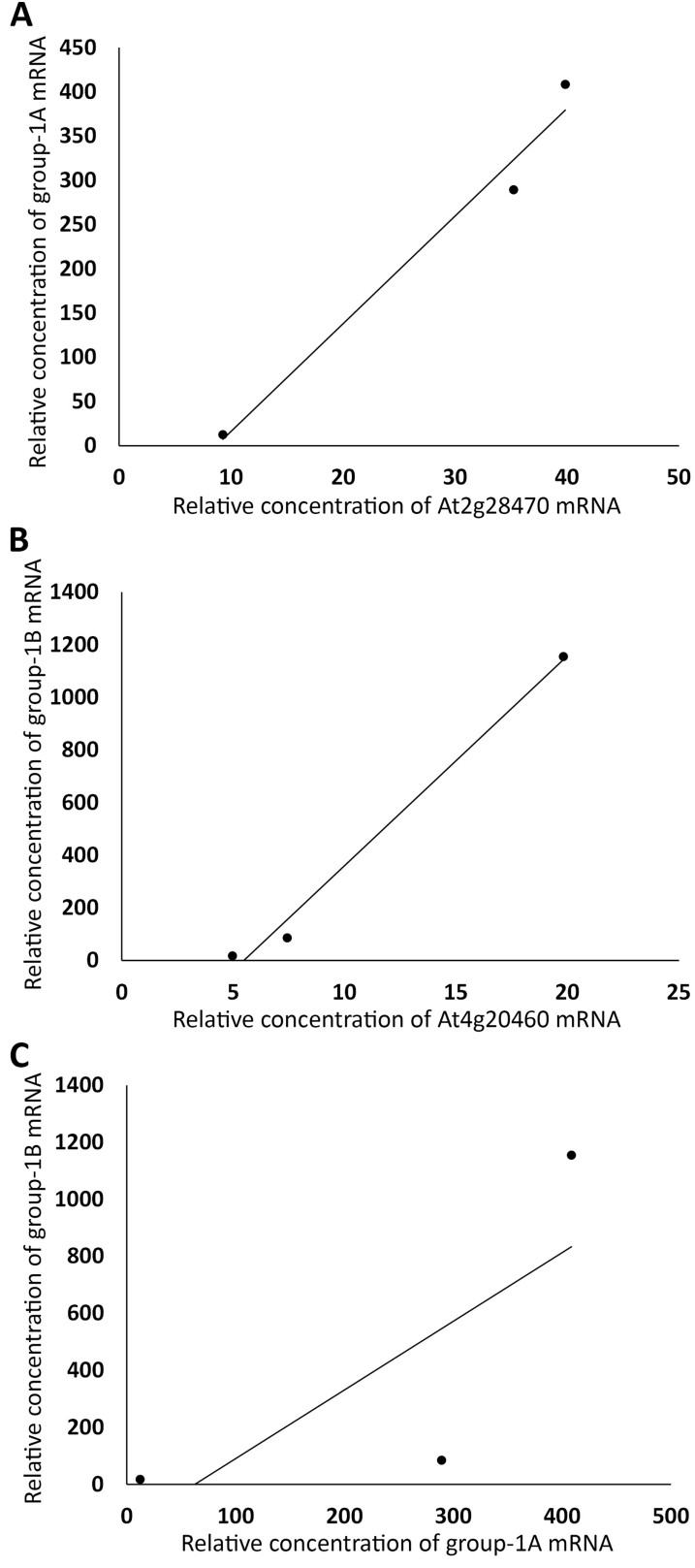

**Figure 3 Grouping auxin-downregulated genes by their linear correlations in relative mRNA concentrations in FUN, CSC, and DSC.** (A) The sums of relative expression levels of all genes in group-1A, excluding those of *At2g28470*, were plotted against those of *At2g28470*. (B) The sums of expression levels

**Figure 3 (continued)**
 of all genes in group-1B, excluding those of *At4g20460*, were plotted against those of *At4g20460*. (C) The sums of relative expression levels of all genes in group-1A, excluding those of *At2g28470*, were plotted against the sums of expression levels of all genes in group-1B, excluding those of *At4g20460*.

RT-PCR, indicating that the insertions likely disrupted the open reading frames of these genes (Fig. 5). Seed germination tests were then conducted for these mutants and Col-0 (wild-type control). The results indicated that both mutant alleles of each locus showed the same trend in germination frequency change; their frequencies were either higher or lower than those of Col-0 (Fig. 6). In particular, mutants of *At1g51170* and *BRL3* had increased germination frequencies and those of *At2g23060*, *HMP39*, and *At4g35060* reduced germination frequencies when compared with their respective controls (paired two-tail *t*-test, $p < 0.05$, $n = 8$ or 12). These results indicated that the AT1G51170 and BRL3 proteins normally negatively affect seed germination and the AT2G23060, HMP39, and AT4G35060 proteins positively affect seed germination. The fact that all tested mutants showed an altered seed germination frequency supports the validity of the approach used in this investigation for predicting genes functioning in seed germination. It is noted here that it was necessary to conduct the seed germination test with paired wild-type and mutant samples, and accordingly, paired *t*-test, because even in the same growth chamber with temperature control, the germination frequencies of Col-0 and the mutants significantly fluctuated between experiments conducted at different times for the same batches of seeds. Such variations, therefore, were likely caused by slight temperature variations in the growth chamber as a result of room temperature variations.

## An integrated gene network for controlling seed germination

Because PYL6 and SNRK2-7 in Table 1 were expected to be involved in ABA signaling and BRL3 in Table 2 was expected to be involved in brassinosteroid signaling, the identified genes seemed to be part of a gene network for seed germination that responds to auxin, ABA, and brassinosteroids. Based on the publicly available information on the genes regulated by these hormones (*Goda et al., 2004*; *Okamoto et al., 2010*; *Tian et al., 2020*; *Liu et al., 2020*), we constructed a gene network that included the six *AFB*s, the 30 genes in Tables 1 and 2, *BES1* and *BZR1* involved in brassinosteroid signaling, and *ABI3, 4,* and *5* involved in ABA signaling (Fig. 7; Fig. S2). The network has interesting features: (1) the genes that are positive for seed germination, including *RBOHB*, *TIP2.2*, and *HMP39* (this investigation), are negatively regulated by both the *AFB1* group and the *AFB5* group. (2) The genes that are negative for seed germination, including *PRX69* and *At4g03140*, are positively regulated by both the *AFB1* group and the *AFB5* group. (3) The *AFB1* group and the *AFB5* group seem to negatively and positively regulate *At2g51170* (inhibiting seed germination, this investigation), *At2g34080* (promoting seed germination), and *At1g78090* (promoting seed germination, this investigation), respectively. (4) Based on how the *AFB1* group and the *AFB5* group are expected to affect the expression of *ABI3, 4,* and *5*, the *AFB1* group and the *AFB5* group seem to negatively and positively regulate ABA signaling,

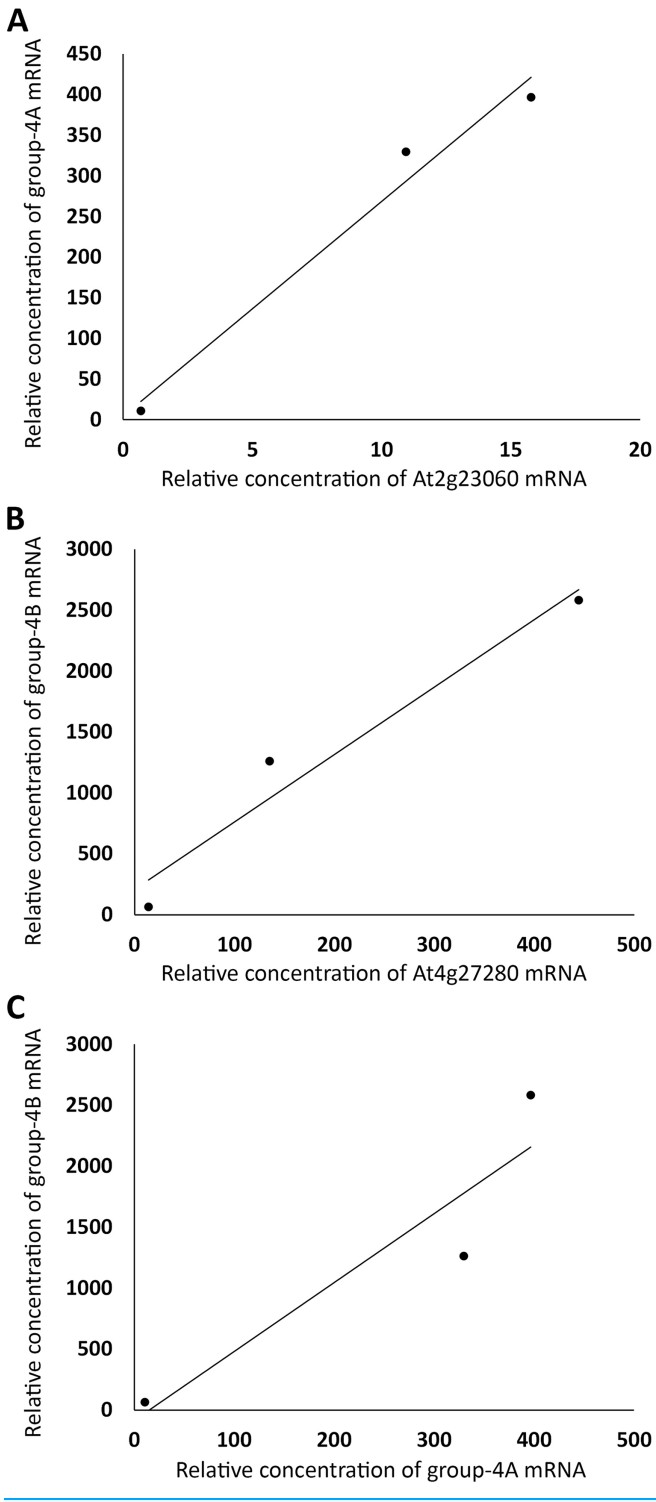

**Figure 4 Grouping auxin-upregulated genes by their linear correlations in relative mRNA concentrations in FUN, CSC, and DSC.** (A) The sums of relative expression levels of all genes in group-4A, excluding those of *At2g23060*, were plotted against those of *At2g23060*. (B) The sums of expression levels of all genes in group-4B, excluding those of *At4g27280*, were plotted against those of *At4g27280*. (C) The sums of relative expression levels of all genes in group-4A, excluding those of *At2g23060*, were plotted against the sums of expression levels of all genes in group-4B, excluding those of *At4g27280*.

**Table 1 Group-1A and -1B genes that are downregulated by auxin and exhibit an up-expression gradient from FUN to CSC.**

| Gene ID | Function of encoded protein |
| --- | --- |
| **Group 1A** | |
| At2G28470[1] | BGAL8, β-galactosidase (*Biswas, 1985*; *Ban et al., 2018*) |
| At4g26320[1] | AGP13, arabinogalactan protein (*Zhang, Held & Showalter, 2020*; *Kaur et al., 2021*) |
| At4g33720 | CAPE3, cysteine-rich secretory proteins |
| At2g40330[2] | PYL6, ABA receptor component (*Kim et al., 2012*; *Takahashi et al., 2020*; *Wang et al., 2024*) |
| At2g15370 | FUT5, fucosyltransferase |
| At5g42180[3] | PRX64, peroxidase (*Jemmat et al., 2020*) |
| At1g09090[1] | RBOHB, respiratory burst oxidase homolog (*Müller et al., 2009*) |
| **Group 1B** | |
| At4g20460[1] | NAD(P)-binding Rossmann-fold superfamily (*Gómez et al., 2006*) |
| At4g35060 | HMP39, heavy metal transport/detoxification |
| At4g40010[2] | SNRK2-7, kinase (*Zhao et al., 2018*; *Lim, Baek & Lee, 2025*) |
| At4g17340[1] | TIP2;2, aquaporin (*Vander Willigen et al., 2006*; *Gattolin, Sorieul & Frigerio, 2011*) |
| At1g15380[3] | GLYI4, ABA and JA crosstalk (*Mei et al., 2023*) |
| At2g44790[2] | UCC2, PLANTACYANIN, light-dependent germination (*Jiang et al., 2021*) |
| At4g26220 | Caffeoyl-coenzyme A O-methyltransferase |
| At1g05260[3] | PER3, cold-inducible cationic peroxidase (*Jemmat et al., 2020*) |
| At1g78090[3] | homologous to trehalose-6-phosphate phosphatases (*Gómez et al., 2010*) |

**Notes:**
[1] Genes that are known or likely to promote seed germination based on published studies.
[2] Genes that are known or likely to inhibit seed germination based on published studies.
[3] Genes that may either promote or inhibit seed germination based on published studies. The remaining genes are unknown for functioning in the seed germination process.

respectively. (5) Outside the aforementioned pathways, the *AFB1* group and the *AFB5* group seem to negatively and positively regulate genes that are positive for seed germination, respectively. These features suggest that this network can produce either a non-germination or a germination outcome, depending on the environmental and/or physiological conditions.

## DISCUSSION

We have constructed a gene network that controls seed germination in Arabidopsis. Signaling processes are broadly represented in this network as it integrates signaling from multiple hormones such as ABA, auxin, brassinosteroids, and possibly cytokinin (*via* ARR7, a regulator in cytokinin signaling) and jasmonic acid (JA) (*via* GLYI4, a protein involved in ABA and JA crosstalk (Tables 1 and 2; Fig. 7)). This network also shows depth as it contains both regulatory proteins and enzymes. Some of the enzymes such as CELLULASE 5, ATXTH22, and the peroxidases may directly act in cell wall degradation or modification during germination. It may be speculated that this network is part of the core structure of the SGN that controls seed germination. Further network studies that expand this network may eventually complete the SGN in Arabidopsis.

**Table 2  Group-4A genes that are upregulated by auxin and exhibit an up-expression gradient from FUN to CSC.**

| Gene ID | Function of encoded protein |
| --- | --- |
| *At5g57560* | XTH22, xyloglucan endotransglucosylase/hydrolase |
| *At5g64100*[2] | PRX69, Class III peroxidase (*Jemmat et al., 2020*) |
| *At1g51170* | Protein kinase, interacts with the transcription factor ATS |
| *At4g30080*[2] | ARF16, AUXIN RESPONSE FACTOR 16 (*Liu et al., 2013*) |
| *At1g19050*[1] | ARR7, regulator in response to cytokinin (*Huang et al., 2017*) |
| *At4g37900*[2] | GRDP2, positive auxin signaling (*Liu et al., 2013*; *Ortega-Amaro et al., 2015*) |
| *At3g13380* | BRL3, brassinosteroid receptor |
| *At2g34080*[1] | Cysteine proteinase (*Martinez et al., 2019*) |
| *At4g03140*[2] | NAD(P)-binding Rossmann-fold superfamily (*González-Guzmán et al., 2002*) |
| *At2g28350*[2] | ARF10, AUXIN RESPONSE FACTOR 10 (*Huang et al., 2017*) |
| *At2g22420*[3] | PRX17, cell wall-localized class III peroxidase (*Jemmat et al., 2020*) |
| *At1g23060* | MDP40, MICROTUBULE DESTABILIZING PROTEIN 40 |
| *At1g22880*[1] | CELLULASE 5 (*Chen et al., 2016*) |
| *At2g23060* | Acyl-CoA N-acyltransferase |

**Notes:**

[1] Genes that are known or likely to promote seed germination based on published studies.
[2] Genes that are known or likely to inhibit seed germination based on published studies.
[3] Genes that may either promote or inhibit seed germination based on published studies. The remaining genes are unknown for functioning in the seed germination process.

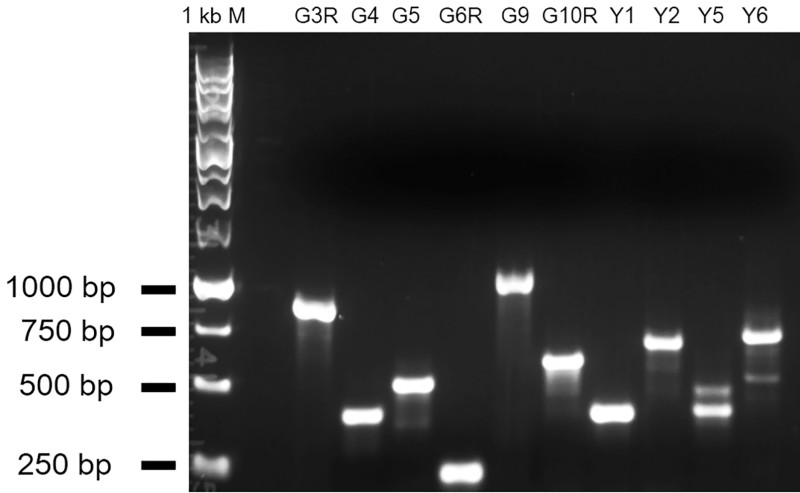

**Figure 5  RT-PCR results showing abnormal transcripts in the five mutants.** The amplified fragments from a gene-specific primer and a T-DNA primer should be plant sequences fused with the T-DNA sequence. The double bands in Y5 and Y6 suggest the existence of alternative splicing of the gene transcript. G3R and G4: mutant alleles of *At1g51170*; G5 and G6R: *brl3* mutant alleles; G9 and G10R: mutant alleles of *At2g23060*; Y1 and Y2: mutant alleles of *At1g78090*; Y5 and Y6: *hmp39* mutant alleles.

The presence of both positive and negative factors in the proposed network likely strikes a balance between maintaining dormancy and initiating germination. Seeds in nature need to maintain dormancy or go into germination according to the developmental and

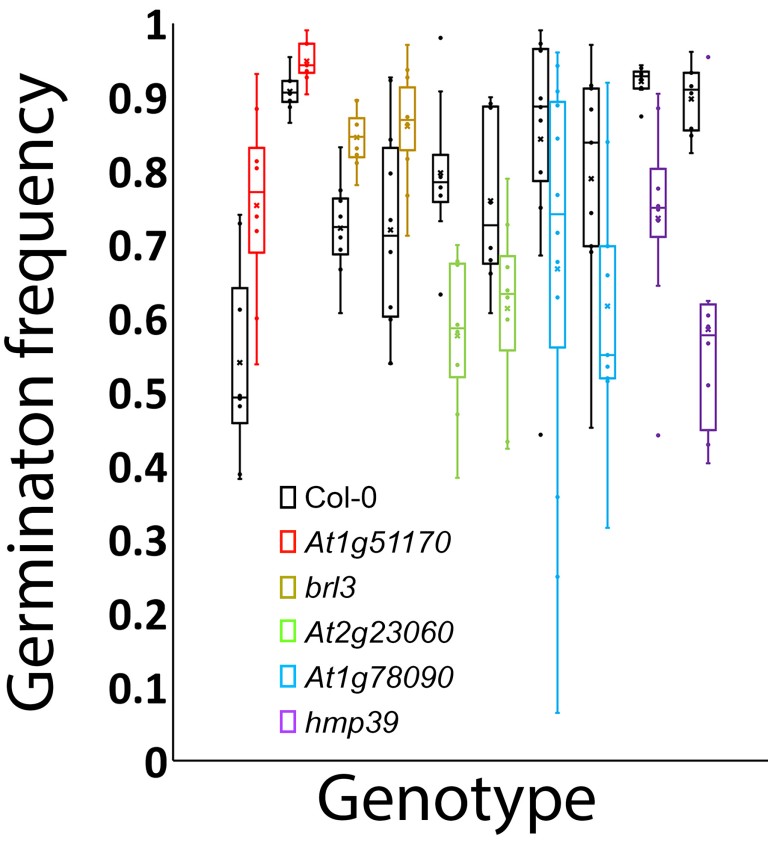

**Figure 6 Germination frequencies of the wild type and 10 mutants.** Black: wild type; red: mutant alleles of *At1g51170*; gold: *brl3* mutant alleles; green: mutants of *At2g23060*; blue: mutants of *At1g78090*; purple: *hmp39* mutants.

environmental conditions. A gene network that perceive multiple signals for regulating both positive and negative factors can conceivably meet such a need. This regulatory mechanism likely leads to the prediction that each of the proteins in the network exerts a small effect on seed germination. Figure 6 indeed shows small effects of individual mutations in the five genes on seed germination. Mechanistically, the small effects may be at least partially explained for the mutants of *BRL3*, *At1g51170*, and *At1g78090* by their nodal positions and connections with other nodes in Fig. 7. For example, on the one hand, *BRL3* suppresses the positive genes *RBOHB*, *TIP2.3*, and *HMP39* and activates the negative genes *ABI4*, which should suppress seed germination. On the other hand, *BRL3* suppresses the negative genes *ABI3 via* both *ARF16/ARF10* and *BES1/BZR1*, which should promote seed germination. In addition, *BRL3* activates negative gene *At1g51170* and positive genes *At2g34080* and *At1g78090 via BES1/BZR1*. These connections between *BRL3* and the downstream genes should result in offsets of positive and negative effects of *BRL3* on seed germination. Therefore, the small effects of the *brl3* mutations on germination observed in this investigation are consistent with the network structure involving *BRL3* in Fig. 7. Similar explanations can be made with the mutations in *At1g51170* and *At1g78090* as the *AFB1* group and the *AFB5* group are expected to negatively and positively regulate them,
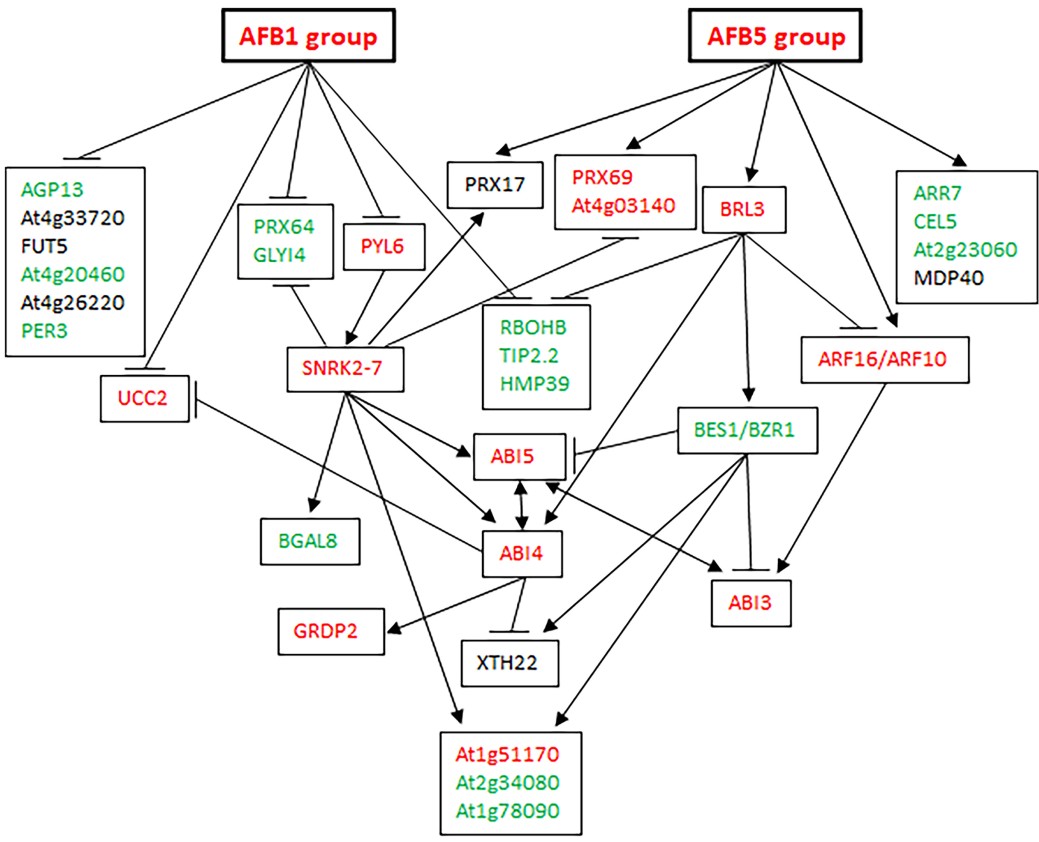

**Figure 7  A gene network affecting seed germination.** Edges ending with an arrow: upregulation; edges ending with a horizontal or vertical bar: downregulation. Green letters: known or likely positive factors for seed germination; red letters: known or likely negative factors for seed germination; black letters: unknown for seed germination.

respectively. Interestingly, both the *AFB1* group and the *AFB5* group negatively regulate the expression of *HMP39* in Fig. 7, and one of its mutations seemed to have the largest effect on seed germination among all the mutants tested (Fig. 6).

The effect of a mutation on seed germination can be expressed as the ratio of the mutant's germination frequency to the corresponding wild type's germination frequency. It will be interesting to find out how such ratios change over different germination conditions. If such ratios are stable or exhibit certain patterns over different germination conditions, a mathematical model may be developed to predict the seed germination outcomes when two or more of the genes in the network are simultaneously altered, which may eventually lead to the prediction of the germination outcomes corresponding to certain parameters of the SGN and environmental conditions. Figure 7 provides a starting framework and candidates for building and testing such a mathematical model.

The identification of the genes in Tables 1 and 2 was based on the key assumption that the *AFBs* are divided into two groups with opposite expression patterns in the FUN-CSC continuum. Our experimental verification of the functions of five previously uncharacterized genes supports the division of the *AFB1* group and the *AFB5* group. Our

previous findings that *AFB1* and *AFB5* have non-overlapping functions and are synergistic in the regulation of seed germination (*Wang et al., 2022*) are consistent with *AFB1* and *AFB5* being in separate *AFB* groups. Interestingly, the gene network in Fig. 7 shows that the primary roles of the *AFB1* group and the *AFB5* group are to down- and upregulate genes, respectively. Further investigation into the establishment and maintenance of separate *AFB* groups in the seed germination and other biological processes may yield new insight into how *AFB*s are coordinated to accomplish one physiological or developmental outcome.

### Funding
This work was supported by a grant (PS21-005) from the Oklahoma Center for the Advancement of Science and Technology to Ming Yang. The funders had no role in study design, data collection and analysis, decision to publish, or preparation of the manuscript.

### Grant Disclosures
The following grant information was disclosed by the authors:
Oklahoma Center for the Advancement of Science and Technology: PS21-005.

### Competing Interests
The authors declare that they have no competing interests.

### Author Contributions
- Ming Yang conceived and designed the experiments, performed the experiments, analyzed the data, prepared figures and/or tables, authored or reviewed drafts of the article, and approved the final draft.
- Yixing Wang performed the experiments, analyzed the data, prepared figures and/or tables, and approved the final draft.

### Data Availability
The raw data and code are available in the Supplemental Files.

### Supplemental Information
Supplemental information for this article can be found online at http://dx.doi.org/10.7717/peerj.19599#supplemental-information.

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
