# Peer review of "Predicting and testing a gene network regulating seed germination in Arabidopsis"

_PeerJ, doi:10.7717/peerj.19599_

## Round 0.1 · original submission · Major Revisions

- More evidence is needed, particularly regarding gene mutations involved in germination regulation, and further validation should be performed.
- A clear hypothesis should be stated, followed by the selection of appropriate experiments to address it. The methodology should be clearly described.

Reviewer 1 ·

Basic reporting

A gene network regulating seed germination in Arabidopsis was predicted in this paper. It is a topic of interest to the researchers in the related areas and the subject fall within the general scope of the journal. However, there are some problems in this paper, such as lack of experimental evidence, confusion between results and discussion, and inaccurate description of methods.

Experimental design

The experimental design is not reliable. To identify genes related to seed germination by analyzing the correlation of gene RNA transcriptional abundance in other experimental results requires other multiple verification experimental evidence, such as electron microscopy observation of seed coat development and western blot experiment of related gene expression.

Validity of the findings

The experimental results belonging to this paper cannot support the main conclusion. The experimental results of this paper are only the seed germination rate of 10 mutants. Validation experimental evidence is incomplete. The proposed regulatory model involves 30 genes, but only up to five genes.

Additional comments

To sum up, the main results of this paper are other people's data, without substantial experimental evidence. The experimental results belonging to this paper cannot support the main conclusion.I recommend rejection

Annotated reviews are not available for download in order to protect the identity of reviewers who chose to remain anonymous.

Reviewer 2 ·

Basic reporting

no comment

Experimental design

no comment

Validity of the findings

no comment

Additional comments

Comment:
This study provides a comprehensive analysis of the regulatory network mediated by auxin-signaling AFB proteins in Arabidopsis seed germination. By integrating transcriptomic data mining with functional validation of novel candidates, the authors establish a gene network model involving crosstalk between auxin, ABA, and brassinosteroid pathways. The work addresses an important gap in understanding maternal control of germination through seed coat signaling. However, there are still some points that need to be clarified and revised:

1. The manuscript did not specify whether the Arabidopsis wild-type and mutant seeds used in germination assays were harvested in the same batch (e.g., synchronized collection timing) or standardized for storage conditions (e.g., temperature fluctuations, relative humidity control). To ensure result reliability, please clarify:
1)Whether all seeds were collected synchronously and stored under uniform conditions (e.g., sealed with desiccants);
2)Whether seeds of different genotypes were stored for identical durations prior to germination testing.

2. The germination assay only provides statistical frequencies (Figure 5), but lacks visual phenotypic evidence. Please provide visual data such as pictures if possible to enhance credibility.

3. Only 5 out of nine candidate genes were selected for mutant validation. Why do not check mutant lines of other genes?

4. Gene expression validation (e.g., qPCR to confirm T-DNA insertion efficiency) is missing and should be supplemented to confirm mutant validity.

5. Minor Revisions
1)“Arabidopsis” should be italicized throughout;
2)Define abbreviations at first mention (e.g., "abscisic acid (ABA)");
3)References should be updated, only two were in recent five years.

Reviewer 3 ·

Basic reporting

This work aims to build a preliminary seed germination network by in silico analysis of publicly available data and its validation with germination experiments.
Authors first determine expression patterns of AUXIN SIGNALING FBOX (AFB) genes in the funiculus (FUN)-chalazal seed coat (CSC) continuum and divide them into two groups as TIR1, AFB1, and AFB4 (down-expression gradient) and AFB2, AFB3, and AFB5 (up-expression gradient). Then they search for genes that correlate with AFB genes and found 118 genes that can be further divided into 4 main and 12 subgroups depending on their expression profiles. Then among these genes they choose five genes that has not been characterized before and test whether those gene affect germination.
In general manuscript is well written. Some sentences can be simplified as they are complex, especially for readers that are not very familiar with the seed germination phenomenon. As the scope of the article and its title is ambitious, I reckon non-specialized readers will also be interested with this work. Examples for such sentences are L54-56, L94-97.

Experimental design

In general methods are explained in detail. However, I have some questions regarding the methods used.

-L127-130 for normalization of microarray data authors use images from Belmont et al. to calculate tissue volumes. Is there any data on size of these tissues on the literature, does your assumption fit with them?

-Why use just filter paper and deionized water for germination assays? In general, a medium that contains plant nutrients such as MS medium is preferred.

-Providing Cytoscape files can be useful for readers that would like to build on your SGN.

-Can you provide criteria used for "manual searches" done during choosing or eliminating genes from different data sets.

Validity of the findings

Figure 3, what three dots represent in the graph? There are more than three genes in this group as listed in supp material.

Figure 5, presentation of data in this graph should be improved showing genotype names on the graph. Moreover, why there is such huge error with At1g78090?

Also, text indicates there is variability for Col-0 but this level of variability (between %55 to %95) indicates that batch of seeds might be heterogenous. Although authors make paired tests this issue leads to doubts about observed results, as result of Col-0 germination would change the outcome of the experiments.

Additional comments

L153, radical should be radicule.

---

## Round 0.2 · Minor Revisions

The reviewers commented that the revised manuscript has potential for publication. However, some minor corrections remain, particularly regarding the statistical analysis, which the authors need to address.

Reviewer 1 ·

Basic reporting

The manuscript is well written. Literature references, sufficient field background/context provided. The structure of the article should conform to an acceptable format .

Experimental design

The experimental design is reasonable.

Validity of the findings

This study provides a comprehensive analysis of the regulatory network mediated by auxin-signaling AFB proteins in Arabidopsis seed germination.

Additional comments

This study provides a comprehensive analysis of the regulatory network mediated by auxin-signaling AFB proteins in Arabidopsis seed germination. However, due to problems in writing, I have doubts about the reliability of the article.
After the author's revision and supplementation, these doubts were eliminated. It is only suggested that when referring to the R2 values related to gene expression correlation in lines 216-246 of the paper, it should be noted whether p is less than 0.05 to comply with the data statistical norms. In addition, for Paired two-tail t-tes in lines 216-246, p should be in lowercase italics.

Annotated reviews are not available for download in order to protect the identity of reviewers who chose to remain anonymous.

Reviewer 2 ·

Basic reporting

no comment

Experimental design

no comment

Validity of the findings

no comment

Additional comments

The authors have addressed all of my concerns. I have no further questions.

---

## Round 0.3 · accepted · Accept

The authors have addressed all the reviewers' comments. After revision, the manuscript has significantly improved and meets the scientific and quality standards for publication.